# Benchmarking Dense and Indiscernible Object Counting with Blueberries

Weihao Bo [* 1]   Yanpeng Sun [* 2]   Jingwen Qin [1]   Fei Shen [3]   Xiaofan Li [4]   Zechao Li [1]

## Abstract

Real-world agricultural counting often operates in the extreme regime of **Dense and Indiscernible Object Counting (DIOC)**, where targets are tiny, clustered, and highly camouflaged. To facilitate research in this domain, we introduce **DIOCblueberry**, a large-scale benchmark that pushes the boundaries of visual perception. Unlike general datasets with salient objects, DIOCblueberry features extreme occlusion and camouflage. Compared to the popular FSC147 benchmark, it contains **1.9× more instances** per image (avg. 108) with an average box pixel ratio that is **7.9× smaller**, serving as a rigorous testbed for model robustness. Standard counting methods struggle in these scenarios due to severe visual ambiguity and scale mismatch. To address this, we propose **MaskCount**, a coarse-to-fine framework that incorporates semantic guidance. MaskCount leverages Vision-Language Models (CLIP) to generate pseudo segmentation masks for background suppression and employs a contrastive loss to maximize feature discriminability between fruits and foliage. Additionally, we design an edge-aware cropping mechanism to resolve boundary truncation in dense clusters. Extensive experiments demonstrate that MaskCount achieves a new state-of-the-art, reducing MAE and RMSE by **49.16%** and **70.50%** respectively on DIOCblueberry, with strong generalization to other agricultural scenes. Our DIOCblueberry benchmark is publicly available at https://huggingface.co/datasets/weihao-bo/DIOCblueberry.

*Equal contribution [1]Nanjing University of Science and Technology, Nanjing, Jiangsu, China [2]Singapore University of Technology and Design, Singapore [3]National University of Singapore, Singapore [4]Zhejiang University, Hangzhou, Zhejiang, China. Correspondence to: Zechao Li <zechao.li@njust.edu.cn>.

*Proceedings of the 43rd International Conference on Machine Learning*, Seoul, South Korea. PMLR 306, 2026. Copyright 2026 by the author(s).

## 1. Introduction

Precision agriculture is fundamentally transforming modern farming by leveraging machine learning to optimize resource allocation and decision-making(Kamilaris & Prenafeta-Boldú, 2018; Eli-Chukwu, 2019; Elavarasan & Vincent, 2020; Van Klompenburg et al., 2020; Bo et al., 2022). Recent multimodal, prompt-based, and parameter-efficient learning methods have further expanded the ability of visual systems to incorporate semantic knowledge and adapt to specialized perception tasks(Bo et al., 2025; 2026; Sun et al., 2022; 2024). Among various tasks, automated fruit counting is critical for yield estimation, harvest planning, and supply chain management(Linker, 2017; Bargoti & Underwood, 2017; Syazwani et al., 2022; Wu et al., 2023). Blueberries, as a high-value global crop, present an urgent demand for such automation. Unlike apples or citrus fruits that are easily discernible, blueberries grow in dense clusters with complex ripening stages. Accurate counting of these fruits in the wild is essential for predicting harvest windows and managing labor, yet it remains an unsolved challenge that bottlenecks the deployment of agricultural robots(Gené-Mola et al., 2020; Afonso et al., 2020; Lyu et al., 2022).

The primary obstacle lies in the extreme visual complexity of blueberry canopies. We define this regime as **Dense and Indiscernible Object Counting (DIOC)**. As shown in Figure 1, DIOC scenarios are characterized by three hostile conditions: **(i)** Micro-scale: The targets are extremely small, often occupying less than 0.24% of the image area; **(ii)** High Density: Fruits appear in tight, overlapping clusters; and **(iii)** Strong Camouflage: Unripe fruits share nearly identical spectral and textural properties with the surrounding foliage. Previous counting methods, which rely on distinct visual boundaries and sufficient inter-object spacing, fail catastrophically in these "low-discriminability" environments.

To advance research in this challenging domain, we introduce **DIOCblueberry**, a large-scale benchmark specifically designed for dense and camouflaged counting in the wild. We constructed this dataset not merely for a specific crop, but to push the boundaries of vision systems in handling ambiguous features. Through rigorous collection and an-

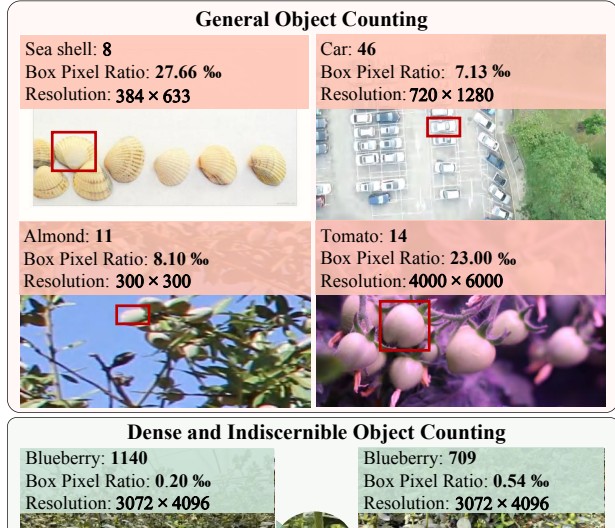

*Figure 1.* Examples of general counting datasets and our DIOCblueberry. Box Pixel Ratio refers to the average pixel area of the bounding boxes relative to the total image area. *Top left*: FSC147, *top right*: CARPK, *middle left*: almond dataset, *middle right*: tomato detection dataset, *bottom*: our DIOCblueberry.

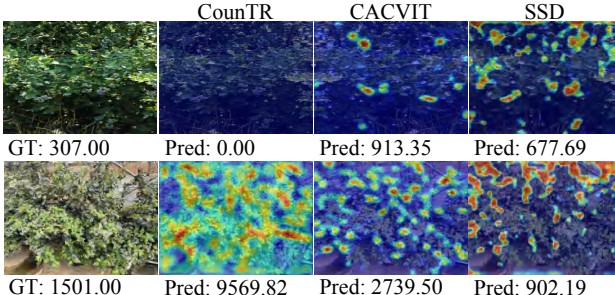

*Figure 2.* Examples of counting results from several state-of-the-art counting methods. *First column*: original images, *second column*: CounTR, *third column*: CACViT, *last column*: SSD.

notation, DIOCblueberry comprises 6,265 high-resolution images with over 679K annotated instances. Compared to existing general benchmarks like FSC147, DIOCblueberry is significantly more demanding. It features an average of 108 instances per image (1.9× higher than FSC147) and an average box pixel ratio of 2.38‰ (7.9× smaller). These statistics confirm that DIOCblueberry represents a *"hard mode"* for counting algorithms, requiring models to distinguish tiny, camouflaged signals from heavy background noise. In addition, the dataset covers diverse field conditions (lighting, occlusion, growth stages) and strong background clutter, which makes it a realistic testbed for deployment.

We evaluate state-of-the-art (SOTA) counting models on this new benchmark and observe substantial performance degradation. Our analysis reveals two critical failures:

- **Visual Ambiguity:** Purely visual features are insufficient to separate green fruits from green leaves, leading to high false positives in background regions.

- **Boundary Loss:** Standard sliding-window inference often truncates dense clusters at patch edges, causing missed counts due to loss of context.

To address these issues, we propose **MaskCount**, a two-stage counting pipeline with semantic background suppression. First, to reduce visual ambiguity, we introduce a CLIP-

based mask module that suppresses cluttered background regions. We use an image-specific background prompt: an LLM proposes candidate background words, and CLIP selects the top-ranked one to guide the mask. Second, to improve feature separation, we employ a contrastive loss in the density regression stage to push apart fruit and background features under strong camouflage. Finally, to handle dense clusters, we design an edge-aware patch cropping mechanism. We use overlapping crops and stitch only valid central regions, reducing boundary truncation and double counting(Wang et al., 2021). Each component targets one key failure mode (background confusion or boundary artifacts) and can be added to existing counting models with minimal changes.

Our approach is simple and effective. Experiments show that MaskCount achieves the best results on DIOCblueberry, reducing MAE and RMSE by 49.16% and 70.50%, respectively. Notably, our edge-aware cropping significantly stabilizes predictions in dense regions, eliminating common edge artifacts. It also improves results on other agricultural datasets, such as almonds and tomatoes. By releasing our code and data, we aim to support future research on dense and indiscernible object counting in real-world environments.

## 2. DIOCblueberry

### 2.1. Image Collection

We collected the images for this study using Xiaomi 13 Ultra and Huawei Mate 60 smartphones. The images were gathered from two regions in China: Yunnan Province and Lianyungang City, Jiangsu Province, both known for blueberry cultivation. The images originates from extensive fieldwork on two large farms (500 acres each), capturing genuine agricultural scenarios essential for yield prediction. Unlike many domains, such images cannot be easily scraped; this real-world grounding is a crucial, difficult aspect of valuable agricultural datasets.

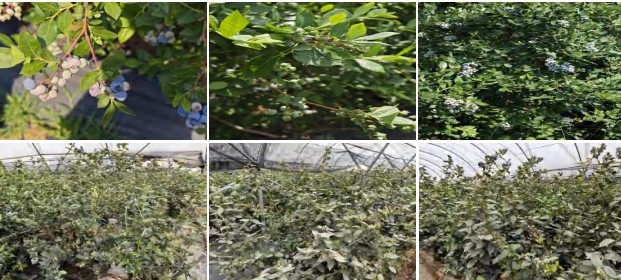

*Figure 3.* Example images from the proposed DIOCBlueberry. *Top left*: less indiscernible and less tiny sample, *top middle*: indiscernible and less tiny sample, *top right*: less indiscernible and tiny sample, *bottom*: indiscernible and tiny samples (typical samples).

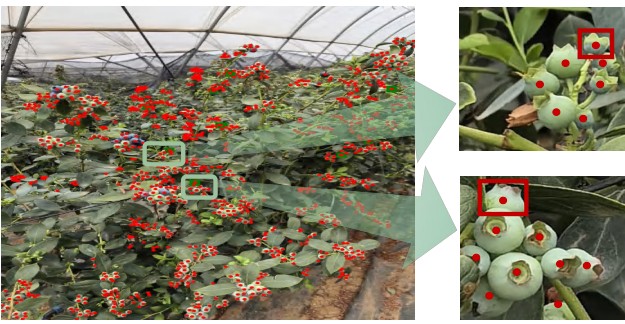

*Figure 4.* Annotation example, with point and box annotations displayed in red.

The images were captured under various lighting conditions, covering the full range of blueberry growth stages, from unripe to fully ripe. This ensures a comprehensive dataset for the DIOC task. Eight professional annotators initially collected a large number of images, after which they carefully reviewed the dataset and removed those that were unsatisfactory or redundant. This process took a total of 300 human hours. The final dataset consists of 6,265 images, some of which are shown in Figure 3. Additional examples of DIOCblueberry images can be found in Appendix A.1.1.

## 2.2. Image Annotation and Analysis

The LabelMe tool was used for annotation. For each image, 3 objects were arbitrarily selected as exemplars, and axis-aligned bounding boxes were drawn for these instances. The remaining objects were annotated with point annotations. In cases of occlusion, an instance was counted and annotated only if less than 90% of it was occluded. While crowd scenes already involve high object density, our scenario poses even greater complexity owing to the severe indistinguishability of objects. This results in substantially more difficult annotation, where each object demands extensive time and multiple rounds of strict validation to achieve accuracy. Figure 4 illustrates an example of image annotation.

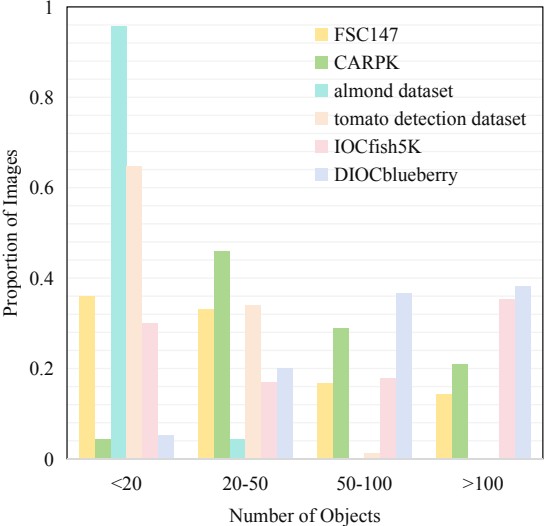

*Figure 5.* Histogram of images distribution across various count ranges.

The annotation process was divided into three stages. First, eight professional annotators were trained to familiarize themselves with their tasks. They were trained on knowledge about blueberry growth and well-annotated samples. They were then asked to annotate 15 images each. The annotations were checked and evaluated. Once the annotator had passed the evaluation, they were allowed to proceed to the next stage. Secondly, the images were distributed to eight annotators, with each annotator being responsible for a portion of the dataset. Annotators were required to discuss confusing cases and reach a consensus. Finally, the annotations were reviewed and refined in two rounds. The second stage required 700 human hours, while the third stage required 350 human hours per round. The total cost of the annotation process amounted to 1,400 human hours.

The dataset consists of 6,265 images, with an average height of 1840 pixels and an average width of 1492 pixels. The dataset contains a total of 679,030 objects, with the maximum number of objects in a single image being 1,980. The dataset is partitioned into a training set of 4,342 images, a validation set of 1,253 images, and a test set of 670 images.

The majority of images in general counting datasets contain fewer than 100 objects. In contrast, a significant proportion of images in our DIOCblueberry dataset contain more than 100 objects, and some even more than 1000. The proportion of images within each object count range across different datasets is provided in Figure 5.

We compare DIOCblueberry with four general counting datasets. FSC147 (Ranjan et al., 2021) is specifically designed for few-shot counting, containing 147 object categories and 6,135 images. CARPK (Hsieh et al., 2017) focuses on vehicle counting in parking lots, with rectangular

*Table 1.* Statistics for existing counting datasets

| Dataset | Images | Avg. Resolution | Count Statistics | | | |
| --- | --- | --- | --- | --- | --- | --- |
| | | | *Total* | *Ave.* | *Max.* | *Avg. Box Pixel Ratio* (‰) |
| FSC147(Ranjan et al., 2021) | 6,135 | 384 × 523 | 344,150 | 56 | 3,701 | 18.76 |
| CARPK(Hsieh et al., 2017) | 1,448 | 720 × 1280 | 89,774 | 62 | 188 | 4.59 |
| Almond dataset(Bargoti & Underwood, 2017) | 620 | 300 × 300 | 4,777 | 8 | 37 | 7.24 |
| Tomato detection dataset(Wu et al., 2023) | 520 | 3406 × 4726 | 9,112 | 18 | 94 | 14.17 |
| DIOCblueberry (Our) | **6,265** | 1840×1492 | **679,030** | 108 | 1,980 | **2.38** |

bounding boxes provided for each vehicle. The ACFR Orchard Fruit Dataset (Bargoti & Underwood, 2017), provided by the agriculture team at the Australian Centre for Field Robotics, The University of Sydney, Australia. It includes apples, mangoes, and almonds, with almond dataset being used for comparison. Tomato detection dataset (Wu et al., 2023) contains images of miniature tomatoes, captured under complex lighting conditions in a plant factory. A visual comparison between DIOCblueberry and other counting datasets is provided in Appendix A.1.2.

Table 1 presents a comparison between our DIOCblueberry dataset and four general counting datasets. DIOCblueberry contains a large number of average object annotations, with the average box pixel ratio being much lower than in general counting datasets. This suggests that the objects are densely distributed and small in size. Additionally, DIOCblueberry exhibits visual ambiguity with the background, which makes the objects harder to distinguish.

In summary, we propose the first specialized dataset for counting dense and indiscernible objects, which is more complex than any existing general counting dataset. Consequently, the substantial human effort dedicated to the challenging on-farm data acquisition and meticulous annotation not only underscores the dataset's complexity, but also matches the scale of labor typically associated with larger benchmarks—highlighting the intrinsic difficulty of curating high-quality datasets tailored for the agricultural DIOC task.

## 3. Proposed Method

We propose MaskCount, a two-stage multi-modal counting method. As shown in Figure 6, the first stage segments objects from backgrounds and generates a background mask to simplify the image for counting. In the second stage, we introduce a contrastive loss to maximize the separation between objects and backgrounds. Additionally, we design an edge-aware patch cropping mechanism that generates overlapping patches to further improve counting accuracy. In the following sections, we detail the architectures of *Crop* and *Stitch* (edge-aware patch cropping mechanism), as well as *Stage 1* (CLIP-based mask generation) and *Stage 2* (estimating density maps with masked images).

### 3.1. *Crop* and *Stitch*: Edge-Aware Patch Cropping Mechanism

General cropping methods result in low counting accuracy at the edges of cropped image patches and noticeable stitching artifacts in the final density map. This is due to the lack of contextual information in edge regions. To address this, we propose an edge-aware patch cropping mechanism that uses a sliding window to generate overlapping image patches. During stitching, only the effective parts of the predicted density maps from image patches are used.

As shown in Figure 6, the high-resolution image size is $(H, W)$. To ensure completeness of the final predicted density map, we add black padding of size $p$ around the high-resolution image, resulting in a padded image of size $(H + 2p, W + 2p)$. The padded image is cropped into $m \times n$ patches using a sliding window of size $(h + 2p, w + 2p)$, where $H = h \times m$ and $W = w \times n$. The horizontal stride of the window is $w$, and the vertical stride is $h$. In other words, overlapping image patches of size $(h + 2p, w + 2p)$ are obtained, with the central region $(h, w)$ being effective. Predictions in the edge regions, with a width of $p$, are discarded, eliminating edge influence and ensuring counting accuracy.

### 3.2. *Stage 1*: CLIP-Based Mask Generation

The primary challenges in DIOC arise from objects being small in size, densely distributed, and exhibiting visual ambiguity with their surroundings. Many real-world applications, such as crop yield estimation, face challenges from highly cluttered backgrounds. These backgrounds include non-object instances like leaves and bushes. Objects with similar colors to backgrounds also contribute to visual ambiguity. Additionally, the small size and dense distribution of the objects complicate subsequent counting tasks. To mitigate these issues, we segment objects from complex backgrounds, thereby reducing the impact of background complexity and easing subsequent counting tasks.

We employ Qwen3 (Yang et al., 2025) to construct a comprehensive list of 121 background classes, mitigating the risk of overlooking complex background attributes common in manual summaries. Given this fixed candidate list, we calculate the similarity between the input image and each background class, selecting the most relevant class as the

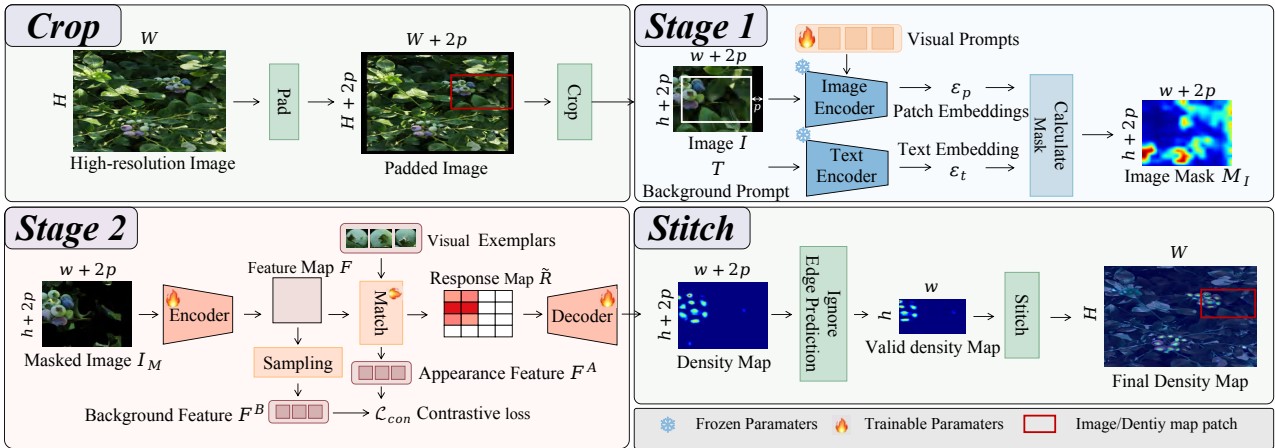

*Figure 6.* Overview of MaskCount. A high-resolution image is cropped into overlapping patches, which serve as the input. The input image **I** is matched with the background prompt **T** at the patch level, activating the successfully matched patches to generate the image mask **M_I**. The masked image is encoded into features **F**. **F** is matched with visual exemplars to generate the response map $\tilde{\mathbf{R}}$, while the appearance features **F^A** of the exemplars are encoded simultaneously. $\tilde{\mathbf{R}}$ is decoded to generate the density map. The valid regions of the density maps are stitched together to produce the final density map. During training, the background feature **F^B** is extracted from **F**, sampled, and used to compute a contrastive loss with **F^A**, aiming to maximize the distance between objects and the background.

background prompt. The prompt used for generation is available in Appendix A.3.

As shown in Figure 6, the input image **I** and background prompt **T** are encoded into patch embeddings $\varepsilon_p$ and text embedding $\varepsilon_t$, respectively. The image and text belong to different modalities, requiring alignment to establish a relationship between them. The cosine similarity map between patch embeddings and the text embedding is calculated to measure the matching degree between image patches and text. Bilinear interpolation is applied to the cosine similarity map, resizing the result to match the input image size, yielding the image mask **M_I**.

While utilizing a pre-trained CLIP backbone (ViT-B/16), we employ a parameter-efficient tuning strategy. The core Vision Transformer and Text Encoder remain frozen to preserve robust vision-language alignment. To adapt for dense counting, we introduce Deep Visual Prompt Tuning (Deep VPT), injecting learnable prompt tokens into multiple transformer layers to capture task-specific inductive biases. Similarly, learnable context vectors are optimized for the text encoder to refine domain-specific representations (e.g., "blueberry"). Beyond these prompts, the projection heads, interaction blocks, and density decoder are fully trainable.

We use InfoNCE loss function (Oord et al., 2018). Specifically, the GT density map is downsampled (via max pooling) to match the CLIP patch resolution. Patches with non-zero density are defined as positive, while background regions are negative. The objective optimizes the learnable text embedding to align with positive object patches while contrasting against the negative set, reinforcing semantic localization.

Minimizing the InfoNCE loss brings positive patch embeddings closer to the text embedding while pushing negative patch embeddings further apart.

$$sum_{p_+} = \sum_{i=0}^{m} \exp(sim(\varepsilon_{p_+}^i, \varepsilon_t)/\tau) \quad (1)$$

$$sum_p = \sum_{j=0}^{m+n} \exp(sim(\varepsilon_p^j, \varepsilon_t)/\tau) \quad (2)$$

$$\mathcal{L}_{stage1} = -\log \frac{sum_{p_+}}{sum_p} \quad (3)$$

where $\{\varepsilon_{p_+}^0, \varepsilon_{p_+}^1, \varepsilon_{p_+}^2, \dots\}$ is the set of positive patch embeddings, $m$ is the number of positive patch embeddings, $\{\varepsilon_p^0, \varepsilon_p^1, \varepsilon_p^2, \dots\}$ is the set of all patch embeddings, $n$ is the number of negative patch embeddings, $\varepsilon_t$ is the text embedding, $sim(\varepsilon_p^i, \varepsilon_t)$ denotes the computation of the cosine similarity matrix between $\varepsilon_p^i$ and $\varepsilon_t$, and $\tau$ is a temperature hyper-parameter.

For the subsequent counting network, we employ a soft masking strategy rather than hard binary thresholding. We generate a continuous mask based on the cosine similarity between image patches and the text embedding. This mask element-wise modulates the visual features, preserving gradient flow and contextual cues around object boundaries—information critical for accurate density estimation.

We provide detailed descriptions of model selection and gradient analysis in Appendix A.4.

### 3.3. *Stage 2*: Estimating Density Map with Masked Image

The input to this stage consists of the backbone feature map $\mathbf{F} \in \mathbb{R}^{B \times C \times H \times W}$ extracted from the masked image, along with exemplar bounding boxes. We distinguish between two exemplar representations based on their information sources. Shape features $\mathbf{F}^S \in \mathbb{R}^{B \times N \times D}$ are explicitly encoded from the geometric properties (width and height) of the exemplar boxes via a multi-layer perceptron (MLP), independent of visual appearance. In contrast, appearance Features $\mathbf{F}^A \in \mathbb{R}^{B \times N \times K^2 \times C}$ capture visual texture and are extracted directly from $\mathbf{F}$ using RoI Align at the exemplar locations with a kernel size of $K \times K$.

As shown in Figure 6, the masked image is encoded into a feature map $\mathbf{F}$. The appearance feature $\mathbf{F}^A$ of visual exemplars is extracted using RoI Pooling. The shape feature $\mathbf{F}^S$ of visual exemplars is extracted using MLP. The feature map, appearance feature, and shape feature undergo cross-attention blocks to extract exemplar prototypes. The process of the cross-attention blocks is described as follows:

$$\mathbf{Q}'_\ell = \text{MHA}\left(\text{LN}\left(\mathbf{Q}_{\ell-1}\right), \mathbf{F}^A, \mathbf{F}^A\right) + \mathbf{Q}_{\ell-1} \quad (4)$$

$$\mathbf{Q}''_\ell = \text{MHA}\left(\text{LN}\left(\mathbf{Q}'_\ell\right), \mathbf{F}, \mathbf{F}\right) + \mathbf{Q}'_\ell \quad (5)$$

$$\mathbf{Q}_\ell = \text{FFN}\left(\text{LN}\left(\mathbf{Q}''_\ell\right)\right) + \mathbf{Q}''_\ell \quad (6)$$

where the inputs at $\ell = 0$ are initialized by the shape feature $\mathbf{Q}_0 = \mathbf{F}^S$, MHA is the standard multi-head attention, LN is layer normalization and FFN is a small feed-forward network. Such a cross-attention blocks structure we used three to get the exemplar prototypes.

Matching process operates in two logical steps. First, features are refined via a cross-attention mechanism, where the shape features $\mathbf{F}^S$ serve as queries to attend to the appearance features $\mathbf{F}^A$ (acting as keys and values) to incorporate exemplar-specific visual contexts. This is followed by attention over the global feature map $\mathbf{F}$ to localize similar instances. Second, the final response map $\tilde{\mathbf{R}}$ is generated via dynamic convolution, where the updated exemplar prototypes $\mathbf{P}$ serve as convolutional kernels applied to the global features: $\tilde{\mathbf{R}} = \mathbf{F} * \mathbf{P}$. Then, the density map is derived by decoding the response map $\tilde{\mathbf{R}}$.

To further increase the separation between the objects and the backgrounds, we apply a contrastive loss. A background mask is derived by thresholding the *Stage 1* prediction (e.g., values $< 0.4$) to identify high-confidence background regions. Feature vectors sampled from $\mathbf{F}$ in these regions constitute the negative sample set, denoted as the background feature $\mathbf{F}^B$. To ensure balanced optimization, we sample a fixed number of background points to match the number of exemplar appearance features (replicating the mean background feature if insufficient pixels are available). Minimizing the contrastive loss between the background

feature $\mathbf{F}^B$ and the appearance feature $\mathbf{F}^A$ increases the separation between objects and backgrounds. The contrastive loss is shown as follows:

$$\mathcal{L}_{con} = \frac{1}{N} \sum_{i=1}^{N} \left(1 - \frac{\mathbf{F}_i^A \cdot \mathbf{F}_i^B}{\|\mathbf{F}_i^A\|_2 \cdot \|\mathbf{F}_i^B\|_2}\right) \quad (7)$$

where $N$ is the number of all possible appearance-background pairs, $\|\cdot\|_2$ denotes the $L_2$ (Euclidean) norm, which is used to normalize the feature vectors. As a result, the fraction computes the cosine similarity between the two vectors, encouraging the optimization to focus on their angular alignment rather than their magnitudes.

We use $\mathcal{L}_{MSE}$ loss function, which measures the $l_2$ difference between the predicted and ground truth density maps. Each cross-attention block generates a density map. The density map produced by the final cross-attention block serves as the output of our model. For a detailed description of the ground-truth density map generation, refer to Appendix A.2.

The final loss is a weighted sum of the two components, with the contrastive loss weight $\lambda_{con}$ controlling their relative contributions. The final loss function is defined as follows:

$$\mathcal{L}_{stage2} = \mathcal{L}_{MSE} + \lambda_{con} \mathcal{L}_{con} \quad (8)$$

We strictly adhere to the standard few-shot counting paradigm. The model does not perform automatic exemplar detection; instead, three randomly selected exemplar bounding boxes are provided per image during both training and inference to specify the target category.

## 4. Experimental Results

In this section, we conduct experiments evaluating our proposed method, MaskCount. We first introduce the evaluation metrics and implementation details, then compare our method with several state-of-the-art methods across different datasets. Finally, we conduct ablation studies to assess the impact of our key designs.

### 4.1. Experimental Settings

**Metrics.** We evaluate performance using two commonly used regression metrics: Mean Absolute Error (MAE) and Root Mean Squared Error (RMSE), which quantify the difference between predicted and ground truth values. MAE reflects the estimation accuracy, while RMSE captures its stability.

**Implementation Details.** All experiments are conducted on 4 NVIDIA H100 GPUs. Performance is evaluated by calculating MAE and RMSE between the model predictions and ground truth values.

*Table 2.* Performance comparison between our method and state-of-the-art methods on different datasets.

| Method | Almond dataset | | Tomato detection dataset | | DIOCblueberry (ours) | | General Crop | |
|---|---|---|---|---|---|---|---|---|
| | *MAE* | *RMSE* | *MAE* | *RMSE* | *MAE* | *RMSE* | *MAE* | *RMSE* |
| CounTR(Liu et al., 2022) | 5.26 | 6.86 | 4.97 | 6.48 | 67.02 | 90.02 | 26.14 | 41.74 |
| LOCA(Đukić et al., 2023) | 2.56 | 3.42 | 2.34 | 3.18 | 7.14 | 14.44 | 5.24 | 7.63 |
| SAFECount(You et al., 2023) | 2.66 | 4.01 | 2.63 | 3.75 | 50.22 | 63.82 | 65.10 | 88.36 |
| CLIP-Count(Jiang et al., 2023) | 5.63 | 6.87 | 7.53 | 9.15 | 15.47 | 29.63 | 58.90 | 72.64 |
| CACViT(Wang et al., 2024) | 5.76 | 7.03 | 5.23 | 6.87 | 26.44 | 37.37 | 19.17 | 23.20 |
| SSD(Xu et al., 2024) | 3.95 | 5.57 | 3.53 | 4.96 | 27.60 | 55.18 | 28.90 | 47.39 |
| **Ours** | **2.03** | **3.02** | **1.91** | **2.71** | **3.63** | **4.26** | - | - |

*Table 3.* Comparison of our method with representative crowd counting, indiscernible object counting, and multi-modal counting methods

| Method | DIOCblueberry | |
|---|---|---|
| | *MAE* | *RMSE* |
| P2PNet(Song et al., 2021) | 8.95 | 28.23 |
| IOCFormer(Sun et al., 2023) | 77.70 | 100.31 |
| CountGD(Amini-Naieni et al., 2024) | 11.35 | 17.74 |
| MaskCount (Ours) | **3.63** | **4.26** |

*Table 4.* Ablation experiments with different combinations of our key designs. VM: vanilla model. *Mask*: CLIP-based mask generation. *Crop*: our edge-aware patch cropping mechanism. *Con*: our contrastive loss.

| Model | DIOCblueberry | |
|---|---|---|
| | MAE | RMSE |
| VM | 7.14 | 14.44 |
| VM+*Mask* | 7.02 | 12.28 |
| VM+*Mask*+*Con* | 6.94 | 10.44 |
| VM+*Mask*+*Con*+*Crop* | **3.63** | **4.26** |

In the first stage, we use the pre-trained CLIP (Radford et al., 2021) model with ViT-B/16 (Dosovitskiy, 2020) as the backbone. The backbone parameters are frozen, while all other parameters are trained on DIOCblueberry training set. We train for 200 epochs using InfoNCE loss, with a batch size of 128 and the AdamW optimizer with a learning rate of $1 \times 10^{-4}$. The entire training process takes approximately 3 hours on 4 NVIDIA H100 GPUs.

In the second stage, the model utilizes the SwAV pre-trained ResNet50 (He et al., 2016) as the backbone. The backbone network parameters are frozen. All other parameters are trained for 60 epochs using the AdamW optimizer, with a learning rate of $1 \times 10^{-4}$ and a weight decay of $1 \times 10^{-4}$. The contrastive loss weight in Eq. (8) is set to $\lambda_{con} = 1 \times e^{-2}$. We train for approximately 1.5 hours on 4 NVIDIA H100 GPUs with a batch size of 2.

We benchmark against few-shot counters to fully leverage the three annotated exemplars provided by DIOCblueberry. Unlike traditional global regressors (e.g., P2PNet), these exemplar-guided models exploit support information to enhance feature extraction. Additionally, they employ state-

of-the-art backbones (e.g., ViT), serving as strong domain-adapted baselines. Consistent inputs of three random exemplars were used across all comparisons.

To ensure a fair comparison, all baselines were fully retrained on the DIOCblueberry dataset following their official protocols. Loss functions were kept identical to the original implementations, predominantly employing Mean Squared Error (MSE) on Gaussian-smoothed ground truth.

During evaluation, the final count for all methods was consistently derived by summing the pixel values of the predicted density maps. Regarding input processing, baselines followed their standard resizing or padding requirements. Crucially, the proposed edge-aware patch cropping mechanism was applied exclusively to our method, aiming to isolate and demonstrate its specific contribution.

*Table 5.* Comparison of different backbones.

| Model | Backbone | DIOCblueberry | |
|---|---|---|---|
| | | *MAE* | *RMSE* |
| SLIP(Mu et al., 2022) | ViT-B/16 | 5.78 | 8.50 |
| SLIP(Mu et al., 2022) | ViT-S/16 | 7.46 | 11.13 |
| CLIP(Radford et al., 2021) | ViT-B/32 | 7.04 | 10.11 |
| CLIP(Radford et al., 2021) | ViT-B/16 | **3.63** | **4.26** |

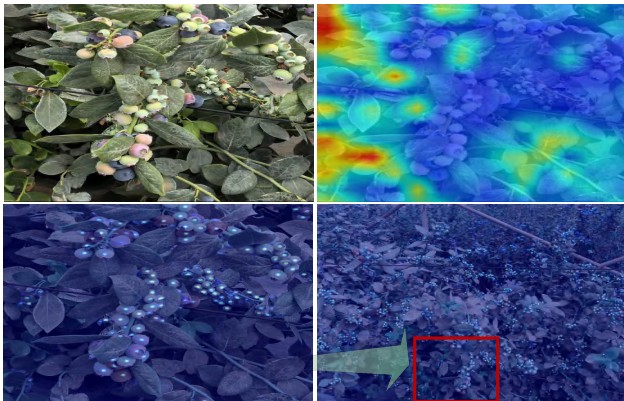

*Figure 7.* Visualization results of our CLIP-based mask generation to segment foreground and background. *Top left*: original image, *top right*: mask, *bottom*: predicted density map.

## 4.2. Comparison with State-of-the-Art Methods

Table 2 presents a comparison of our method with several state-of-the-art methods on DIOCblueberry and two general counting datasets. As shown in the table, our method outperforms previous counting methods on the challenging DIOCblueberry dataset and achieves state-of-the-art performance on both almond dataset and tomato detection dataset. Furthermore, Table 2 details the performance under general cropping conditions. Even in this setting, our method maintains a significant lead, attributed to its robust multimodal capabilities, thereby further demonstrating its effectiveness.

Due to the small size of blueberry fruits, the two-stage method CounTR (Liu et al., 2022) struggles to count objects effectively on DIOCblueberry. In contrast, our two-stage multi-modal method, MaskCount, drastically improves performance, reducing MAE from 67.02 to 3.63 and RMSE from 90.02 to 4.26. When compared to the vision-language pretraining-based method CLIP-Count (Jiang et al., 2023), MaskCount achieves a notable reduction in MAE by 76.54% and RMSE by 85.62%. Additionally, MaskCount surpasses ViT-based CACViT (Wang et al., 2024) and ResNet-based models: LOCA (Đukić et al., 2023) , SAFECount (You et al., 2023), and SSD (Xu et al., 2024).

*Table 6.* Comparison of different cropping methods.

| Method | DIOCblueberry | |
| --- | --- | --- |
| | *MAE* | *RMSE* |
| Resize | 6.94 | 10.44 |
| General crop | 6.36 | 9.67 |
| Our crop | **3.63** | **4.26** |

We also evaluate the crowd counting method P2PNet(Song et al., 2021), the indiscernible object counting method IOC-Former(Sun et al., 2023), and the state-of-the-art multi-modal counting method CountGD(Amini-Naieni et al., 2024) on DIOCblueberry. Table 3 presents a comparison of counting performance across these methods. IOC-Former represents the current best performance on IOC-fish5K—the largest existing dataset for indiscernible object counting—while P2PNet is a representative crowd counting approach. CountGD leverages multi-modal information to further improve counting accuracy. Despite these strong baselines, our method consistently achieves the best performance, demonstrating its robustness and effectiveness. These results also underscore the inherent difficulty of the agricultural DIOC scenario, which poses significant challenges beyond those in existing datasets.

## 4.3. Ablation Study

We conduct a comprehensive ablation study to illustrate the contributions of our design components: CLIP-based mask generation, the edge-aware patch cropping mechanism, and the contrastive loss.

The results presented in Table 4 demonstrate that each component of our design contributes to performance improvement, confirming the effectiveness of every design. Specifically, *Mask* leads to performance improvements, with MAE and RMSE decreasing by approximately 1.68% and 14.96%, respectively.

We compare the effects of using CLIP and SLIP (Mu et al., 2022) for object and background segmentation on counting performance. As shown in Table 5, the model achieves optimal counting performance when using the pre-trained CLIP model with ViT-B/16 as the backbone. Figure 7 presents the visual results of our CLIP-based mask generation. In addition, we compare the counting performance under different prompts in Appendix A.5.

In Table 6, our edge-aware patch cropping mechanism improves performance, reducing MAE and RMSE by approximately 47.69% and 59.20%, respectively. Furthermore, our edge-aware patch cropping mechanism outperforms the general cropping method by reducing MAE from 6.36 to 3.63 and RMSE from 9.67 to 4.26. The results for general crop and details on the selection of padding size for our crop are provided in Appendix A.5.

As shown in Figure 8, the image on the *left* shows the counting results using general cropping. The blueberry in the red circle is located at the edge of the cropped image patch and is counted twice. On the *right*, the counting results using our edge-aware patch cropping mechanism are shown, where the same blueberry is counted only once. These results demonstrate that our method effectively eliminates stitching artifacts by disregarding edge predictions.

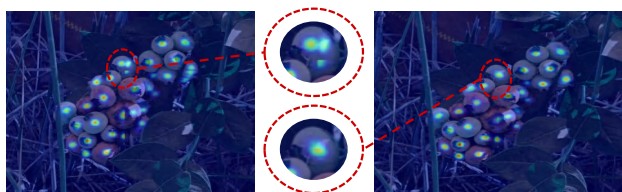

*Figure 8.* Final counting results visualization of general cropping compared to ours. *Left*: general cropping method, *right*: our edge-aware patch cropping mechanism. The red circle marks a blueberry located in the edges of the cropped image patches.

*Table 7.* Comparison of different contrastive losses.

| Loss | DIOCblueberry | |
| --- | --- | --- |
| | *MAE* | *RMSE* |
| Sigmoid-based Similarity Loss | 6.63 | 10.22 |
| InfoNCE Loss | 5.20 | 7.85 |
| Our contrastive loss | **3.63** | **4.26** |

As shown in Table 7, we compare the performance of different contrastive losses. The sigmoid-based loss suppresses feature similarity between target and background via a logistic function but lacks explicit angular constraints, leading

to limited discrimination in dense scenes. InfoNCE introduces relative contrastive learning, however, its reliance on batch-wise normalization and clear positive–negative correspondence is less suitable for low visual separability scenarios. In contrast, our contrastive loss directly enforces angular separation between target and background features without temperature scaling or complex sampling strategies. It leads to a reduction in MAE by approximately 30.19% and RMSE by nearly 45.73%. More visual results are provided in Appendix A.6. In addition, the analysis of our limitations is provided in Appendix A.7.

## 5. Conclusion

In this work, we formally define the task of Dense and Indiscernible Object Counting (DIOC) and introduce DIOCblueberry, a challenging large-scale benchmark characterized by tiny, clustered, and highly camouflaged targets. To overcome the limitations of existing counters in these extreme regimes, we propose MaskCount, a multi-modal framework that integrates CLIP-based semantic background suppression with a novel pixel-level contrastive loss to resolve visual ambiguity. Furthermore, we introduce an edge-aware patch cropping mechanism that effectively mitigates boundary artifacts common in high-resolution processing. Extensive experiments demonstrate that MaskCount establishes a new state-of-the-art on DIOCblueberry and exhibits strong generalization across diverse agricultural scenarios. We hope this benchmark and methodology will facilitate future research into robust counting systems for complex, real-world environments.

## Acknowledgements

This work was supported by National Natural Science Foundation of China (Grant No. 62425603) and Basic Research Program of Jiangsu Province (Grant No. BK20240011). We also thank Jiangsu Zhishou Ecological Agriculture Co., Ltd. for providing the raw collected data.

## Impact Statement

This paper significantly advances precision agriculture by addressing the critical bottleneck of automated counting in complex, unstructured environments. By enabling accurate yield estimation and harvest planning for dense and camouflaged crops, our work contributes to optimized resource allocation and labor management in modern farming. Furthermore, the proposed dataset and methodology serve as a rigorous testbed for robust visual perception, facilitating the broader deployment of agricultural robotics and intelligent monitoring systems.

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

# A. Appendix

## A.1. Dataset

### A.1.1. DIOCBLUEBERRY

Figure 9 shows more examples of DIOCblueberry, with *(a)* depicting less dense images and *(b)* depicting dense images. From left to right in *(a)*, the object count ranges are ≤20, 20–50, and 50–100. While in *(b)*, the object count ranges are 100–500, 500–1000, and >1000. This demonstrates that our DIOCblueberry includes a diverse set of images, covering varying densities of objects from sparse to dense. It effectively showcases the diversity in object distribution and is capable of handling counting tasks across different object densities. This demonstrates the generalization ability of our method.

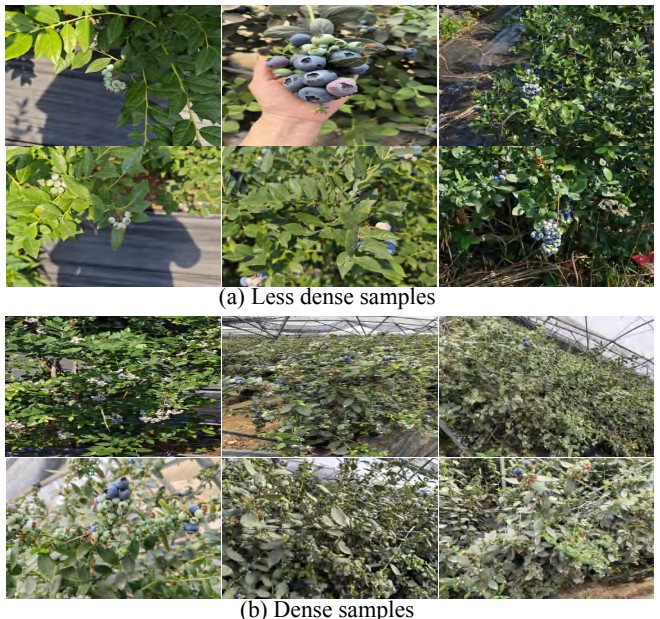

(a) Less dense samples

(b) Dense samples

*Figure 9.* More examples of our DIOCBlueberry.

### A.1.2. COMPARISON WITH OTHER DATASETS

Figure 10 presents a visual comparison between DIOCblueberry and general counting datasets. Compared to all these datasets, our scenes are significantly more complex, featuring indiscernible objects and higher object densities. These characteristics make the counting task in DIOC considerably more challenging and representative of real-world deployment scenarios.

Figure 11 provides a visual comparison between DIOCblueberry and IOCfish5K, the largest indiscernible object counting dataset. While IOCfish5K contains visually ambiguous objects, DIOCblueberry introduces additional challenges. The objects exhibit high visual similarity to the background, scenes are significantly more cluttered, and occlusions are more severe. These factors collectively make DIOCblueberry a more complex and demanding benchmark for evaluating counting performance in real-world scenarios.

As shown especially in the bottom row, although IOCfish5K also contains a large number of objects, they are densely clustered in localized regions. In contrast, our objects are more widely dispersed across the scene, making the counting task in our scenes more challenging.

## A.2. Ground Truth Density Map Generation

As our DIOCBlueberry employs point and box annotations, we should transform these discrete points into continuous density maps. For density map generation, images and point annotations are first rescaled to the target resolution, and a sparse impulse map is initialized at the rescaled point locations. Gaussian filtering is then applied to diffuse the discrete points into a continuous distribution: when reference boxes are available, the kernel size is adaptively determined by the average

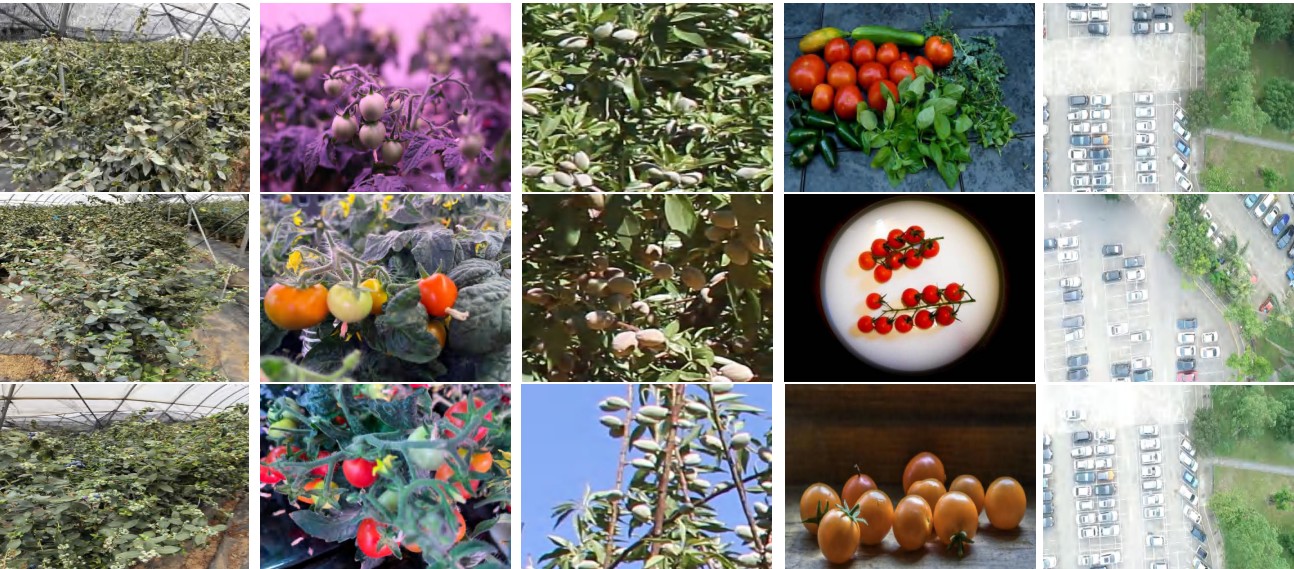

*Figure 10.* Visual comparison between DIOCblueberry and general counting datasets. *The first column:* our DIOCBlueberry, *the second column:* tomato detection dataset, *the third column:* almond dataset, *the fourth column:* FSC147, *the fifth column:* CARPK.

box scale; otherwise, a fixed kernel is used. This process preserves the spatial distribution of annotations while producing smooth and scale-consistent density representations that serve as continuous supervisory signals for model training.

The generation of ground-truth density maps follows a geometry-adaptive Gaussian kernel strategy derived from the point annotations and exemplar bounding boxes. First, for an image with $N$ annotated objects, we initialize a binary delta map where pixels at the annotated point locations are set to 1, and all others are 0. To determine the spread ($\sigma$) of the Gaussian kernel, we utilize the 3 exemplar bounding boxes provided in the DIOCblueberry annotations. Specifically, the standard deviation $\sigma$ for the Gaussian filter is dynamically set to one-eighth of the average dimensions ($\bar{w}, \bar{h}$) of these exemplar boxes, formulated as $\sigma = (\bar{w}, \bar{h})/8$. This geometry-adaptive sizing ensures that the density accumulation regions match the approximate scale of the objects in that specific image. In rare cases where exemplar boxes are unavailable, we employ a fixed kernel size with $\sigma = 4$ pixels, which is an empirical fallback value chosen to match the average object scale in the dataset to prevent training instability. Finally, the ground truth density map is generated by convolving the delta map with this Gaussian kernel. This smoothing operation maintains the total energy of the map, ensuring that the spatial integral of the density map precisely equals the total object count (i.e., $\sum D_{gt} \approx N$), thereby providing a spatially continuous supervision signal for the regression loss.

### A.3. Qwen Input for Background Descriptor Generation

To generate candidate background descriptors, we employ Qwen3 with a fixed text-only input designed to summarize the background of dense and visually indiscernible blueberry counting scenarios. The input instructs the model to produce a list of single-word descriptors, primarily nouns, covering as much non-target area as possible, and to output only the words separated by commas without any additional text: *"summarize the background in in-field blueberry farm images for dense and indiscernible object counting. The background excludes blueberry fruits and may include leaves, branches, stems, soil, sky, grass, weeds, shadows, sunlight glare, trellis, irrigation pipes, nets, mulch film, plastic bags, tags, hands, tools, or other farm objects. Output a comma-separated list of single-word descriptors (primarily nouns, up to 150 words). Output only the word list."*

The resulting candidate list of 121 descriptors is then used in subsequent image-specific similarity ranking to select the most relevant background descriptors for each image.

### A.4. Model Selection and Gradient Analysis

We adopt CLIP rather than the Segment Anything Model (SAM) as the backbone of Stage 1 for two main reasons. First, CLIP is explicitly designed to align visual representations with textual semantics. In dense counting tasks, accurately

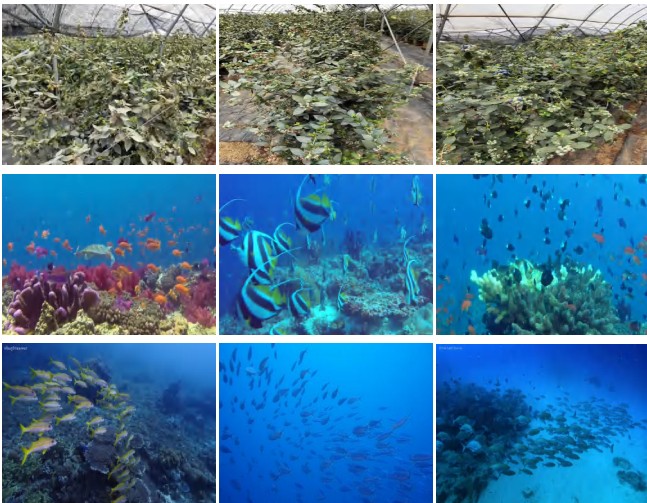

*Figure 11.* Visual comparison between DIOCblueberry and IOCfish5K. *Top:* our DIOCBlueberry, *middle and bottom:* IOCfish5K.

distinguishing the target object (e.g., "blueberry") from visually similar distractors (e.g., leaves or shadows) requires strong semantic discrimination, which CLIP naturally provides through text prompting. Second, SAM primarily relies on geometric prompts (e.g., points or bounding boxes) and low-level texture boundaries. In dense counting scenarios such as DIOC, target objects are often extremely small and exhibit blurred or ambiguous boundaries, making it difficult for SAM to produce reliable instance masks. In contrast, CLIP's patch-level semantic representations are more robust to such conditions and enable more stable localization through cross-modal similarity.

For the contrastive loss $\mathcal{L}_{stage1}$ 3, taking the derivative with respect to the similarity score $s_k = \text{sim}(\varepsilon_p^k, \varepsilon_t)/\tau$ yields: for a positive sample $k$, $\frac{\partial \mathcal{L}}{\partial s_k} = -p_k^{\text{pos}} + p_k^{\text{all}}$, where $p_k^{\text{pos}} = \frac{e^{s_k}}{\sum_{i \in \mathcal{P}} e^{s_i}}$ and $p_k^{\text{all}} = \frac{e^{s_k}}{\sum_{j \in \mathcal{P} \cup \mathcal{N}} e^{s_j}}$. For a negative sample $k$, $\frac{\partial \mathcal{L}}{\partial s_k} = p_k^{\text{all}}$.

This indicates that the loss encourages the entire positive set to dominate the softmax distribution, with the intra-positive distribution serving as the ideal target. Compared to the conventional single-positive InfoNCE, this formulation distributes gradients across multiple positives, reducing training variance and instability. Additionally, it robustly aggregates intra-class diversity and mitigates the effect of false negatives. And it imposes consistency constraints over the whole positive set, thereby learning more compact and discriminative feature representations.

### A.5. Ablation study

Table 8 shows the counting results corresponding to CLIP-based mask generation using different prompts in first stage. We conduct experiment using class name as prompt to identify foreground objects for counting. However, results show that this approach underperforms using background-descriptive prompt to mask out irrelevant regions before counting. This highlights the effectiveness of background suppression over foreground guidance in DIOC scenes.

*Table 8.* Ablation experiments with different background prompts of our CLIP in the first stage. *Top 1*: the most similar text selected from the candidate texts, *top 3*: top three most similar texts selected from the the candidate texts

| Prompt type | Prompt | DIOCblueberry | |
| --- | --- | --- | --- |
| | | MAE | RMSE |
| foreground | "blueberry" | 7.60 | 11.85 |
| | "leaf" | 5.42 | 8.21 |
| background | *top1* (ours) | **3.63** | **4.26** |
| | *top3* | 8.50 | 11.31 |

*Table 9.* Ablation experiments with different padding sizes of our edge-aware patch cropping mechanism on DIOCblueberry

| Padding size | DIOCblueberry | |
| --- | --- | --- |
| | MAE | RMSE |
| 16 | 6.05 | 9.95 |
| 32 (ours) | **3.63** | **4.26** |
| 64 | 6.84 | 11.71 |

The experiments on background prompt selection demonstrate that using the highest-ranked prompt performs better than using just "leaf" or the top three prompts. These results indicate that selecting the most relevant background prompt for mask generation can significantly improve counting accuracy. By choosing the most relevant prompt, we can more effectively separate the objects from the background, thereby enhancing the overall performance of DIOC task.

Table 9 shows the results of our edge-aware patch cropping mechanism with different padding sizes, indicating that a padding size of 32 leads to a lower counting MAE and RMSE. This suggests that using a padding size of 32 effectively minimizes the interference from edge effects, improving the accuracy of bounding box positioning. As a result, our model is able to make more accurate predictions of object counts.

### A.6. Subjective Performances

Our CLIP-based mask generation alleviates the counting challenges posed by complex backgrounds. It segments the objects and background, which improves the quality of the final density maps and leads to better object counting performance. The heatmaps and the segmentation results of our CLIP-based mask generation are shown in Figure 12, in the heatmaps, the background areas are activated. In the visualization images, the background areas are masked. In the first four rows, we visualize the mask results for four different object densities in DIOCblueberry, showing that our CLIP-based mask generation can accurately segment the objects and background, whether the objects are sparse or dense. As shown in the last two rows in Figure 12, we display the visualized counting results for two general counting datasets: the almond dataset and tomato detection dataset. The results demonstrates that our CLIP-based mask generation performs effectively on general counting datasets as well. Our CLIP-based mask generation effectively segments the objects and background, reducing the background complexity and lowering the difficulty of the subsequent object counting task.

The results of CounTR (Liu et al., 2022), LOCA (Đukić et al., 2023), SAFECount (You et al., 2023), CLIP-Count (Jiang et al., 2023), CACViT(Wang et al., 2024), SSD (Xu et al., 2024) and ours on our DIOCBlueberry, almond dataset, and tomato detection dataset are shown in Figure 13. In our DIOCblueberry, the object count per image ranges from 10 to 2000. In the first four columns, we visualize the counting results of four different images with ground truth counts ranging from 10 to around 2000. The results demonstrate that our method, MaskCount, can handle both sparse and dense counting scenarios. As shown in the last two rows in Figure 13, our method is able to count the objects that are small in size, dense distribution, and visual ambiguity with their surroundings in complex backgrounds. It demonstrates superior performance on DIOCBlueberry compared to state-of-the-art methods. The last two columns of Figure 13 show the visualized counting results on general counting datasets: the almond dataset and tomato detection dataset. These results show that our MaskCount also performs excellently on general counting datasets, demonstrating the versatility of our method.

### A.7. Limitation analysis

While MaskCount achieves significant improvements in counting accuracy, its inference speed and computational cost are higher compared to one-stage methods such as LOCA and SSD. This suggests a trade-off between accuracy and efficiency. However, this limitation is not fundamental and can be addressed. As a promising direction for future work, we plan to distill MaskCount into a lightweight, end-to-end model to better meet the demands of real-world deployment scenarios.

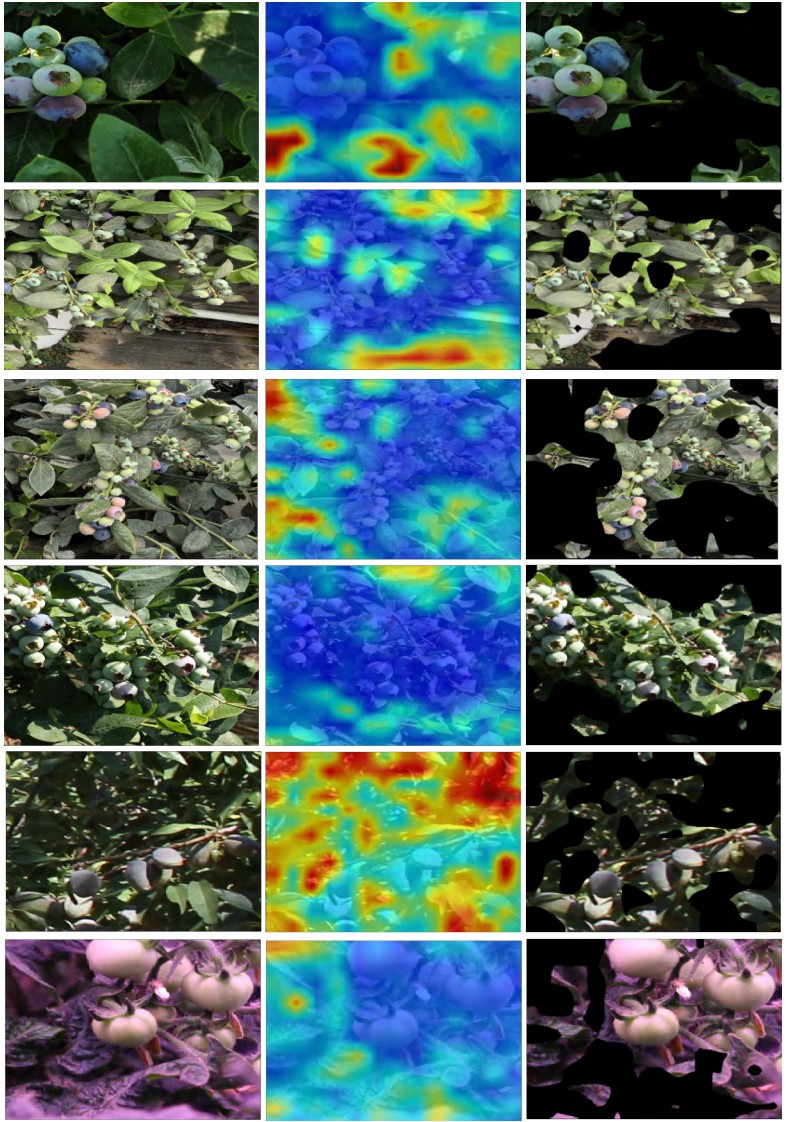

*Figure 12.* Visualization results of our CLIP-based mask generation to segment foreground and background. *The first four rows*: our DIOCBlueberry, *the fifth row*: the almond dataset, *the last row*: tomato detection dataset.

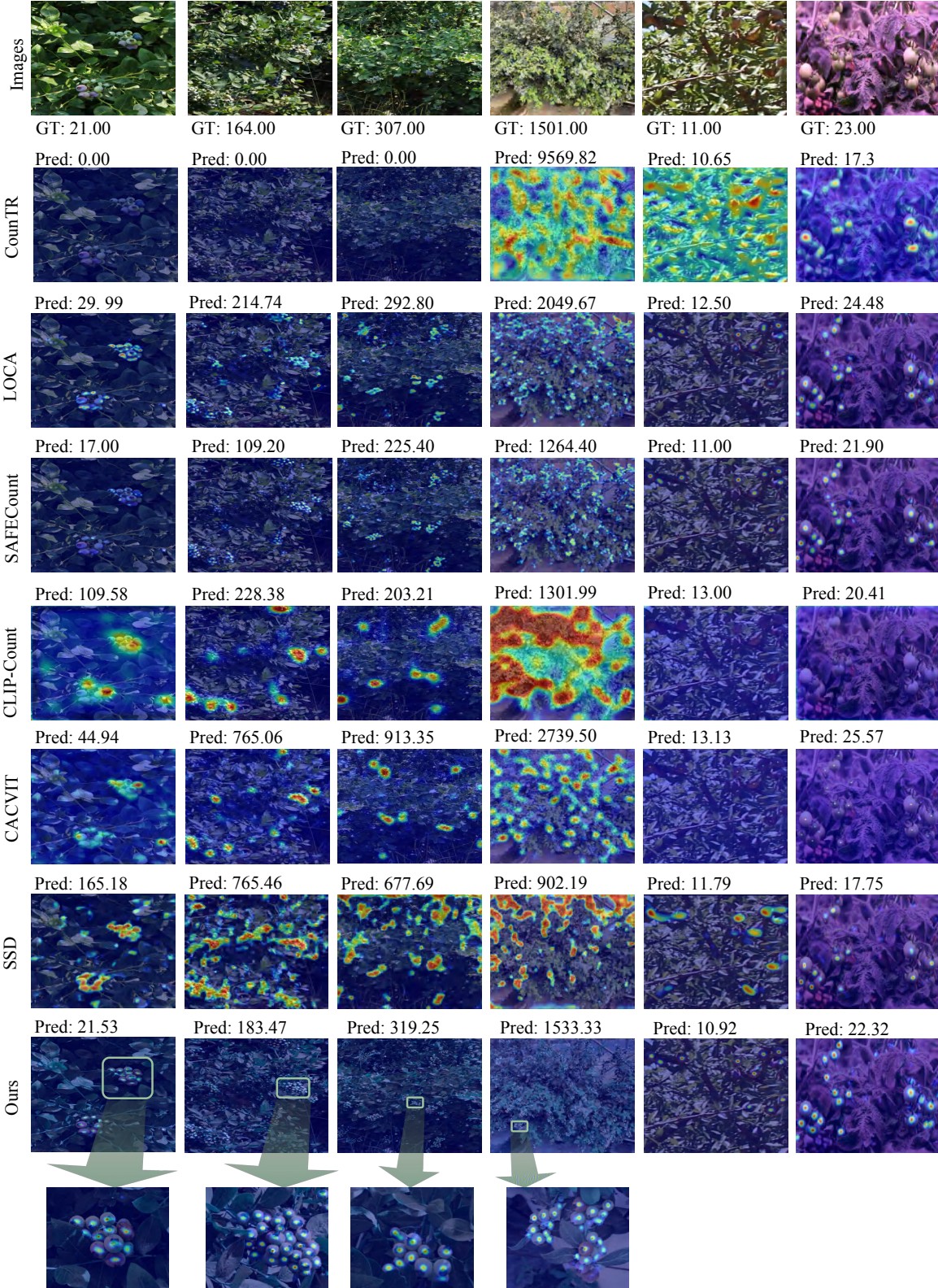

*Figure 13.* Qualitative results on our DIOCBlueberry, almond dataset, and tomato detection dataset. *The first four columns:* our DIOCBlueberry, *the fifth column:* almond dataset, *the sixth column:* tomato detection dataset (bottom).

