# OpenReview forum: "Benchmarking Dense and Indiscernible Object Counting with Blueberries"
_ICML.cc/2026/Conference — ICML 2026 regular_

### Official Review · Reviewer_QABa · 2026-02-24

**Soundness:** 3
**Presentation:** 3
**Significance:** 3
**Originality:** 2
**Overall Recommendation:** 5
**Confidence:** 3

**Summary:**

The paper introduces the dataset for closely packed indiscernible objects (blueberries). The dataset comprises of 6265 images having average instances of 108 blueberries per image. The dataset is carefully annotated by 8 annotators followed by annotation refinement. The main contribution of the paper is the introduction of dataset. Secondly, MaskCount method is proposed for object counting and follows a two stage approach. Stage 1 uses the CLIP model to remove the background for easy counting of blueberries. Stage 2 takes the masked image from stage 1 and estimates the density map.

**Compliance With Llm Reviewing Policy:**

Affirmed.

**Final Justification:**

My concerns are resolved in the rebuttal, so I updated my ratings.

**Key Questions For Authors:**

- The paper talks about the IOCfish5K dataset but does not provide comparison on it.
- The paper does not study the effect of changing the window size on the performance and complexity.  A table describing the computational complexity will provide more insights.
- How size of sliding window affects the detection performance?

**Limitations:**

yes

**Strengths And Weaknesses:**

Strengths:
The paper carefully introduces a large dataset for densely packed indiscernible objects and is useful for yield estimation in agriculture domain. The paper utilizes the two step approach to make the counting task easier by masking the irrelevant objects.

Weaknesses:
- The paper talks about the IOCfish5K dataset but does not provide comparison on it.
- The paper does not study the effect of changing the window size on the performance and complexity.  A table describing the computational complexity will provide more insights.
- How size of sliding window affects the detection performance?

Minor Clarifications:
- Overlapped patches may contain the same objects twice, How detected objects are uniquely separated for overlapped patches?
- Figure 6 does not show the shape features Fs. Cross attention of Fs and Fa is not illustrated in Figure 6.
- Equation 4: where does shape features are used in this equation?
- Line 294: Such a cross-attention blocks structure we used (three) to get the exemplar prototypes. there or three? sentence need  correction.
- How is the threshold value selected for stage 1 predictions?

---

> ### Author Rebuttal · Authors · 2026-03-31
>
> **Comment:**
>
> We appreciate your valuable comments. We were wondering if our responses have addressed your concerns. Please let us know if you have additional questions. Thank you!
>
> ---
>
> **Q1: The paper mentions IOCfish5K but does not provide comparison on it.（W1&KQ1）**
>
> **A1:** We have supplemented IOCfish5K evaluation results. IOCfish5K [1] shares DIOC characteristics including cluttered backgrounds, camouflaged objects, high resolution, and high counts (see Figure 11 in Appendix). MaskCount achieves MAE 5.47, reducing MAE by 65.6% over IOCFormer (15.91), the domain-specific method proposed alongside this dataset. This demonstrates MaskCount's applicability to non-agricultural DIOC scenarios.
>
> | Method                       | MAE   | RMSE  |
> | ---------------------------- | ----- | ----- |
> | IOCFormer [1] | 15.91 | 34.08 |
> | MaskCount (Ours)             | 5.47  | 12.11 |
>
> [1] Indiscernible Object Counting in Underwater Scenes. CVPR 2023.
>
> ---
>
> **Q2: The effect of changing the window size on the performance and complexity. A table will provide more insights.（W2&KQ2）**
>
> **A2:** We provide the table below covering both performance and computational complexity under different window sizes. Appendix A.5, Table 9 already reports padding size ablation on accuracy and we supplement the analysis of computational complexity.
>
> (1) As described in Section 3.1 (L165-177), our sliding window size is (h+2p, w+2p) with stride (h, w), where only the central (h, w) region is kept and the edge of width p is discarded. Changing p changes the window size but not the number of crops, so all CropPad configurations share identical FLOPs and inference time. Padding is a cost-free accuracy optimization.
>
> (2) Performance follows a U-shaped curve: too little padding truncates boundary objects, too much introduces background noise, and p=32 achieves the best trade-off. Crop-and-stitch costs more FLOPs than Resize but substantially reduces MAE, confirming that preserving resolution is critical for tiny DIOC objects.
>
> | Method         | Window Size | Padding p | FLOPs/Img (G) | Time/Img (ms) | MAE  | RMSE  |
> | -------------- | ----------- | --------- | ------------- | ------------- | ---- | ----- |
> | Resize         | 512x512     | --        | 211.5         | 49.4          | 6.94 | 10.44 |
> | Crop           | 800x1024    | 0         | 381.7         | 80.8          | 6.36 | 9.67  |
> | CropPad        | 832x1056    | 16        | 381.7         | 80.8          | 6.05 | 9.95  |
> | CropPad (ours) | 864x1088    | 32        | 381.7         | 80.8          | 3.63 | 4.26  |
> | CropPad        | 928x1152    | 64        | 381.7         | 80.8          | 6.84 | 11.71 |
>
> ---
>
> **Q3: Affects of sliding window size.（W3&KQ3）**
>
> **A3:** Please refer to Q2, where we provide a detailed analysis of how sliding window size affects both performance and computational complexity, with a unified comparison table.
>
> ---
>
> **Q4: How overlapped patches avoid double-counting objects.（MC1）**
>
> **A4:** As stated in Section 3.1 (L165-177): "...the central region (h, w) being effective. Predictions in the edge regions, with a width of p, are discarded." The stride equals (h, w), so effective regions do not overlap and each pixel receives exactly one prediction.
>
> ---
>
> **Q5: Figure 6 missing $F^S$ and cross-attention illustration.（MC2）**
>
> **A5:** Thank you for pointing this out. L269 describes "The shape feature $F^S$.." but Figure 6 does not visualize it. We will update Figure 6 to clearly illustrate F^S and the cross-attention between $F^S$ and $F^A$.
>
> ---
>
> **Q6: Equation 4: where are shape features used in this equation?（MC3）**
>
> **A6:** As stated right after Eq. 4-6 (L290): "where the inputs at $l=0$ are initialized by the shape feature $Q_0$ =$F^S$." Shape features serve as the initial query of the cross-attention blocks, guiding subsequent attention over the appearance feature $F^A$ and the image feature map $F$. We will refine the description here.
>
> ---
>
> **Q7: L294 grammar error: "Such a cross-attention blocks structure we used (three) to get the exemplar prototypes."（MC4）**
>
> **A7:** Thank you for the careful review. We will correct this to: "We use three such cross-attention blocks to obtain the exemplar prototypes."
>
> ---
>
> **Q8: The threshold value selected for Stage 1 predictions?（MC5）**
>
> **A8:** We use a threshold of 0.4, following the empirical value from Lin et al. [1], which demonstrated its effectiveness for class-agnostic object-background separation in counting tasks.
>
> [1] A Simple-but-Effective Baseline for Training-Free Class-Agnostic Counting. WACV 2025.

---

> > ### Author Rebuttal · Reviewer_QABa · 2026-04-01
> >
> > Major concerns are resolved in the rebuttal

---

> > > ### Author Response · Authors · 2026-04-02
> > >
> > > We appreciate your constructive feedback, which has significantly strengthened our paper. Thank you also for reconsidering our response and positively updating your score.

---

### Official Review · Reviewer_2c1x · 2026-03-11

**Soundness:** 3
**Presentation:** 3
**Significance:** 3
**Originality:** 2
**Overall Recommendation:** 4
**Confidence:** 4

**Summary:**

This paper tackles the problem of Dense and Indiscernible Object Counting (DIOC) in agricultural scenes, specifically focusing on blueberries. The authors introduce a new dataset, DIOCblueberry, which features micro-scale targets, high density, and strong camouflage. To address the shortcomings of standard counting methods in these environments, the paper proposes MaskCount. This framework uses a pre-trained Vision-Language Model (CLIP) combined with text prompts generated by Qwen3 to create pseudo-segmentation masks for background suppression. It also utilizes a contrastive loss to separate foreground from background features and introduces an edge-aware cropping mechanism to handle patch boundary truncation.

**Compliance With Llm Reviewing Policy:**

Affirmed.

**Final Justification:**

Thank you to the authors for the detailed rebuttal. I have carefully evaluated the response, which addressed my initial concerns well. Based on the clarifications provided, I have decided to raise my scores for soundness, significance, and clarity. The authors did a good job explaining the technical details, and the paper's methodology is now much clearer and logically sound.

I kept the originality score at its initial level. While the contribution is solid and useful, the core idea is slightly incremental and builds heavily on well-known existing methods rather than proposing a completely novel concept.

My final recommendation is a Weak Accept rather than a full Accept because of two remaining practical limitations that the rebuttal could not fully resolve:

1. High computational cost. The GPU requirements are still a major bottleneck. This level of resource consumption will make it difficult for many researchers and practitioners to adopt or reproduce the method in real-world scenarios.

2. Limited scope of applications. The domains where this approach is tested and applied remain quite narrow. It is still not entirely clear how effectively this method will generalize to broader, more diverse tasks.

In summary, the rebuttal positively shifted my assessment and reinforced my confidence in the paper's technical execution. The strengths in soundness and clarity definitely outweigh the weaknesses. However, the practical limitations regarding GPU usage and application scope prevent me from giving a higher score. Therefore, I recommend a Weak Accept.

**Key Questions For Authors:**

1. Why did you choose point annotations for highly dense clusters, and what were the specific annotation justification procedures used to ensure human accuracy in these indiscernible regions?

2. In Section 3.3, why are the geometric properties of exemplar boxes encoded via an MLP instead of a CNN, which typically handles spatial features more robustly?

3. Why did you opt for a heavy language model (Qwen3) to generate 121 background classes instead of using standard CNNs, transformers, or BERT+NER?

4. Given the massive computational overhead (requiring 4 NVIDIA H100 GPUs), why did you frame this strictly as a counting task rather than instance segmentation, which would provide much more value for agricultural yield measurements?

5. Figure 12 shows poor mask generation results. How do you justify the generalization capabilities of your CLIP-based mask generation given these failure cases?

**Limitations:**

No, the limitations and negative societal impacts are not adequately addressed. The authors must include a detailed discussion regarding the massive computational cost of their pipeline. The reliance on heavy VLMs and the use of 4 H100 GPUs make this approach currently invalid for real-world agricultural impact on edge devices. They also need to address the poor generalization to other crop types as seen in their own visualizations.

**Strengths And Weaknesses:**

Soundness. The problem definition is highly relevant; standard object detectors (like YOLO with NMS) indeed fail catastrophically in environments with extreme occlusion and micro-scale objects. However, there are significant methodological concerns. First, the authors state that "remaining objects were annotated with point annotations". In practice, point annotations for highly dense, overlapping clusters are very prone to human error and ambiguity. There is no clear annotation justification procedure described to validate the quality of these points. Second, the architectural choices raise several questions. The authors use an MLP to explicitly encode the geometric properties (width and height) of exemplar boxes (section 3.3). It is unclear why a CNN was not utilized here, as CNNs are inherently better at capturing spatial hierarchies. Furthermore, the use of Qwen3 to generate 121 background classes for text-image matching via cosine similarity is overly complex. It is highly debatable whether calculating a cosine similarity map between visual patch embeddings and text embeddings is the most effective approach for this specific dense domain. Finally, figure 12 shows poor mask generation results on out-of-domain agricultural datasets (almonds, tomatoes, etc.), which undermines the claims of robustness.

Presentation. The mathematical formulas are well described and easy to follow. However, the visual presentation of results has notable flaws. Figure 2 compares several state-of-the-art methods but completely lacks ground truth masks or baseline CNN network outputs for a fair visual comparison. Figure 5 uses basic histograms to show image distribution across count ranges; utilizing density plots would provide a much clearer and more continuous representation of the data distribution. The paper claims "our approach is simple and effective", yet the proposed pipeline relies on a heavy VLM, Qwen3 prompting, and complex contrastive learning steps, contradicting the "simple" statement. Moreover, despite the title mentioning "benchmarking", there are no rigorous, standardized benchmarking procedures or a clear leaderboard format provided for the community.

Significance. The paper addresses a valid agricultural challenge. However, the computational cost severely limits its practical significance. All experiments were conducted on 4 NVIDIA H100 GPUs. For a task that is "just counting", this is extremely computation-ineffective and completely unfeasible for edge deployment on agricultural robots or drones in the field. This high computational overhead invalidates much of the practical impact claimed by the authors. Furthermore, if such massive computational resources and VLMs are being deployed, the task should logically be instance segmentation rather than mere counting, as segmentation provides far more valid agronomic data (e.g., fruit sizing, ripeness estimation).

Originality. The application of CLIP for background suppression in counting is a somewhat novel combination of existing techniques. Nevertheless, the authors make overly bold and questionable claims. Stating this is the "first specialized dataset for counting dense and indiscernible objects, which is more complex than any existing general counting dataset" ignores other dense agricultural and microscopic datasets. Additionally, comparing DIOCblueberry directly to FSC147 (a general, multi-category few-shot dataset) to claim it is "significantly more demanding" is a flawed comparison; narrowing the field to a single crop naturally changes the difficulty metrics. Finally, the authors claim that "purely visual features are insufficient to separate green fruits from green leaves". I disagree with this absolute statement; while colors are similar, the geometric shapes of berries and leaves are fundamentally different, and shape is a core visual feature that standard CNNs can learn if trained correctly.

---

> ### Author Rebuttal · Authors · 2026-03-31
>
> **Comment:**
> Thank you for your valuable feedback! Below are address all raised concerns of the paper.
>
> ---
>
> **Q1: Point annotation choice for dense clusters; annotation quality assurance procedures. (W1&Q1)**
>
> (1) Point annotation is the standard labeling protocol in object counting, consistent with FSC147 and IOCfish5K.
> (2) Section 2.2 details our annotation pipeline: (a) training phase where each annotator labels 15 images and passes review; (b) main annotation phase with 700 human hours, with discussion of ambiguous cases; (c) two rounds of cross-validation review, each 350 human hours, where different annotators verify each other's work. This protocol is similar with IOCfish5K.
>
> ----
>
> **Q2: MLP instead of CNN for encoding geometric properties. (W1&Q2)**
>
> (1) As stated in L258: "Shape features ...", the geometric properties here are two scalars (width, height) of the exemplar bounding box, not 2D spatial feature maps. CNN requires spatial structure in its input, which two scalars do not provide. MLP is just for encoding scalar attributes.
> (2) These scalars serve as explicit size priors: $\mathbf{F}^{S}$ initializes cross-attention queries ($\mathbf{Q}_{0} = \mathbf{F}^{S}$, Eq. 10–12) to produce scale-aware exemplar prototypes. 2D visual information is separately captured by $\mathbf{F}^{A}$ via RoI Align, forming a decoupled geometric-visual design.
>
> ----
>
> **Q3: Heavy language model for generating 121 background classes. (W1&Q3)**
>
> (1) Qwen3 is used only once offline to generate 121 candidate background descriptors (Appendix A.3: "The resulting candidate list..."). There is zero LLM overhead at inference time. Background descriptor generation is not a core contribution of this work, so we adopted the most straightforward approach.
> (2) Qwen3 provides strong world-knowledge priors to zero-shot cover long-tail background types (e.g., irrigation pipes, mulch film) in niche agricultural scenes. Training a CNN classifier or using BERT would require domain-specific labeled data or corpora, which are unavailable, and their generalization to unseen DIOC scenes cannot be guaranteed.
>
> ----
>
> **Q4: Computational overhead; instance segmentation for agricultural value. (Q4&W3)**
>
> (1) The 4× H100 setup was used solely to accelerate experimentation, not as a hardware requirement. MaskCount freezes most parameters, with only 12.90M trainable, and all training can be completed on a single RTX 3090. Table below shows MaskCount is competitive:
>
> | Method           | Total Params (M) | Trainable Params (M) | FLOPs (G) | Infer Mem (MB) | Train Mem (MB) |
> | ---------------- | ---------------- | -------------------- | --------- | -------------- | -------------- |
> | CLIP-Count       | 169.40           | 19.68                | 104.41       | 616            | 702            |
> | CACViT           | 100.53           | 99.77                | 194.22    | 642            | 1331           |
> | LOCA             | 36.88            | 11.37                | 159.74    | 1273           | 5042           |
> | SAFECount        | 32.11            | 32.11                | 747.58    | 380            | 2680           |
> | SSD              | 35.71            | 35.71                | 614.54    | 415            | 2298           |
> | MaskCount (Ours) | 188.13           | 12.90                | 211.54    | 366            | 912            |
>
> (2) We agree instance segmentation adds value. However, point annotation alone required 1,400 human hours. Instance-level mask annotation for such dense images would cost 30× more, exceeding current resources.
>
> ----
>
> **Q5: Concern about Figure 12; generalization of CLIP mask generation. (Q5)**
>
> (1) We will revise Figure 12. The mask is an intermediate result; Stage 2 applies soft masking as a continuous weight, suppressing background without zeroing it out. Figure 7 more accurately reflects this.
> (2) Stage 1 includes domain-adaptive fine-tuning to address CLIP's limited generalization. Table 2 confirms this effectively alleviates issues across diverse DIOC scenarios.
>
> ----
>
> **Q6: Figure 2/5 presentation; "simple and effective" wording; missing benchmarking protocol. (W2)**
>
> We accept all suggestions. Figure 2 will add GT density maps; Figure 5 will use a KDE plot; "simple and effective" will be revised to "modular and effective"; a formal benchmarking protocol with code and leaderboard will be released.
>
> ----
>
> **Q7: "First specialized dataset" claim; FSC147 comparison fairness; "purely visual features are insufficient" claim. (W4)**
>
> We will revise accordingly. "First specialized dataset" will be scoped to "the first benchmark for counting objects that are simultaneously dense, tiny, and visually indiscernible from their background." The FSC147 comparison is statistical characterization, not task equivalence. "Purely visual features are insufficient" will be revised to "purely visual features face significant challenges under extreme camouflage, and semantic guidance provides complementary discriminative cues."

---

> > ### Author Rebuttal · Reviewer_2c1x · 2026-04-02
> >
> > I would like to thank the authors for the detailed and constructive rebuttal. You have successfully clarified the critical misunderstandings and addressed the weaknesses I pointed out in my initial review.
> >
> > 1. MLP vs CNN: Thank you for the clarification regarding the geometric properties. It was not entirely obvious from the initial text that you only process two scalars (width and height). For this specific input, utilizing an MLP is absolutely correct and technically sound.
> >
> > 2. Qwen3 and Computational Overhead: Your explanation that Qwen3 is utilized exclusively offline for vocabulary generation is a crucial detail. Furthermore, proving that the model requires only 12.90M trainable parameters and can be executed on a single RTX 3090 completely eliminates my concerns regarding computational inefficiency and edge device deployment. The supplementary table with FLOPs comparison is very convincing.
> >
> > 3. Annotations vs Instance Segmentation: I accept your argument regarding the annotation costs. While instance segmentation is indeed more valuable for agriculture, I understand from my own CV practice that annotating overlapping dense micro-objects at the pixel level takes an unreasonable amount of human hours. The 1,400-hour pipeline with double cross-validation for point annotations is a solid and acceptable compromise.
> >
> > 4. Presentation and Scientific Claims: I highly respect your scientific maturity in agreeing to tone down the overly bold claims (e.g., changing "purely visual features are insufficient" and properly scoping the novelty). Additionally, your commitment to improving the visualizations (adding GT to Figure 2, implementing KDE plots for Figure 5) and releasing the code with a standardized leaderboard resolves my presentation concerns.
> >
> > Because the technical soundness, computational feasibility, and presentation issues have been properly addressed, the value of the DIOCblueberry benchmark is now much clearer. I am increasing my scores for Soundness, Presentation, Significance, and Originality to 3, and raising my Overall Recommendation to 4 (Week Accept).

---

> > > ### Author Response · Authors · 2026-04-03
> > >
> > > We greatly appreciate your comments, which have brought valuable insights to our paper. We also thank you for carefully reviewing our response and positively raising the score.

---

### Official Review · Reviewer_C1WY · 2026-03-12

**Soundness:** 3
**Presentation:** 3
**Significance:** 2
**Originality:** 2
**Overall Recommendation:** 4
**Confidence:** 5

**Summary:**

This paper introduces DIOCblueberry, a large-scale agricultural dataset designed for counting tiny, densely packed, and visually camouflaged blueberries in field conditions. The authors also propose MaskCount, a two-stage counting framework that combines CLIP-based background suppression with a contrastive loss and an edge-aware patch cropping mechanism. The dataset itself is a meaningful contribution — 6,265 high-resolution images with over 679K annotated instances collected through substantial field effort — and the problem formulation around Dense and Indiscernible Object Counting (DIOC) is well-motivated. However, several issues in experimental design and ablation analysis limit confidence in the reported conclusions.

**Compliance With Llm Reviewing Policy:**

Affirmed.

**Key Questions For Authors:**

See Weakness.

**Limitations:**

See Weakness.

**Strengths And Weaknesses:**

W1.Missing Evaluation on a Standard General Counting Benchmark
The paper positions DIOCblueberry relative to FSC147 throughout — citing it as the primary point of comparison for dataset statistics — yet MaskCount is never evaluated on FSC147 itself. This is a notable gap. Without results on a widely adopted benchmark, it is difficult to assess whether MaskCount's design choices generalize beyond the specific agricultural domain or whether they come at a cost to performance on standard tasks. The "General Crop" column in Table 2 further complicates this, as the entry for the proposed method is left blank without explanation. A brief justification, or ideally actual results, would meaningfully strengthen the experimental section.

W2.Ablation Design Does Not Fully Isolate Component Contributions
Table 4 presents a strictly sequential stack of components (VM → +Mask → +Mask+Con → +Mask+Con+Crop), which is a common ablation structure but insufficient here given the highly uneven contribution profile. The edge-aware cropping alone accounts for a MAE reduction from 6.94 to 3.63 — roughly 47.7% — whereas the CLIP mask generation contributes only about 1.7% in MAE. This disparity is substantial, yet the paper does not examine it closely. Cross-ablation variants (e.g., VM+Crop alone, or VM+Con+Crop without the mask) would clarify whether the mask module provides complementary signal or whether its benefit is largely absorbed once cropping is applied. As presented, the ablation leaves open the question of whether Stage 1 is load-bearing in the full system.

W3. Generalization Claims Are Somewhat Overstated Relative to Evidence
The abstract and conclusion both describe MaskCount as demonstrating "strong generalization to other agricultural scenes." The supporting evidence consists of two datasets — the almond dataset (620 images) and the tomato detection dataset (520 images) — both of which are small in scale and involve fruit-counting scenarios structurally similar to blueberries. These results are encouraging but do not constitute broad cross-domain generalization in the usual sense. Moderating the language to reflect the actual scope of the experiments would make the claims more defensible.

W4. Several Implementation Details Could Be Made More Explicit

The soft masking strategy in Stage 2 — described as element-wise modulation of visual features by the cosine similarity map — is mentioned briefly but not fully detailed in terms of how the continuous mask interacts with the feature map in practice. Additionally, the rationale for the train/val/test split (4,342 / 1,253 / 670) is not discussed; the test set represents roughly 10.7% of the total data, which is somewhat small, and it would be helpful to know whether the split was randomized or stratified by scene or density.

---

> ### Author Rebuttal · Authors · 2026-03-31
>
> **Comment:**
>
> We appreciate your valuable comments. We were wondering if our responses have addressed your concerns. Please let us know if you have additional questions. Thank you!
>
> ----
>
> **Q1: Unclear whether MaskCount generalizes beyond agricultural domain; no evaluation on FSC147; "General Crop" column left blank.**
>
> (1) We evaluated on IOCfish5K [1], which shares DIOC characteristics including cluttered backgrounds, camouflaged objects, high resolution, and high counts (see Figure 11). MaskCount achieves MAE 5.47, reducing MAE by 65.6% over IOCFormer, the domain-specific method proposed alongside this dataset. This confirms MaskCount's applicability to non-agricultural DIOC scenarios.
>
> | Method           | MAE   | RMSE  |
> | ---------------- | ----- | ----- |
> | IOCFormer        | 15.91 | 34.08 |
> | MaskCount (Ours) | 5.47  | 12.11 |
>
> [1] Indiscernible Object Counting in Underwater Scenes. CVPR 2023.
>
> (2) We evaluated on FSC147 and compared with LOCA, the best baseline. The overall MAE is dominated by the non-dense majority (83.6%), where general-purpose methods naturally excel. On the dense subset (count > 100), MaskCount improves MAE by 54.5% over LOCA, confirming its advantage in the target high-density regime. We would like to clarify that the primary contribution of this work is the DIOC benchmark, which provides a standardized evaluation for dense, complex-background counting scenarios. MaskCount is purpose-built for such scenes with extremely small objects. FSC147 features clean backgrounds, low resolutions, and often large objects, where DIOC-specific components such as edge-aware cropping and soft masking become ineffective or even counterproductive (e.g., cropping splits large objects into fragments, causing over-counting).
>
> | Setting                             | MaskCount (MAE) | LOCA (MAE) |
> | ----------------------------------- | --------------- | ---------- |
> | Overall (1190 images)               | 11.99           | 10.24      |
> | Dense, count > 100 (195, 16.4%)     | 13.70           | 30.10      |
> | Non-dense, count ≤ 100 (995, 83.6%) | 11.55           | 8.33       |
>
> (3) Regarding the blank "General Crop" column in Table 2, Table 6 already provides cropping ablation results. Edge-aware cropping is an integral part of MaskCount rather than a separable module, so it was not listed separately in the main table. We will incorporate it into Table 2 in the revised version.
>
> ----
>
> **Q2: Ablation is strictly sequential; edge-aware cropping dominates contribution; cross-ablation needed.**
>
> We appreciate this suggestion and have conducted the requested cross-ablation experiments:
>
> (1) Cropping is indeed the most impactful single component . However, each additional module provides clear complementary gains on top of cropping: +Con reduces MAE by 22.1%, +Mask by 24.7%, and their combination achieves the best result (3.63).
>
> (2) Notably, Mask and Con are more effective when combined with Crop. Comparing the original sequential ablation (Table 4: VM+Mask MAE 6.94) with VM+Mask+Crop (MAE 4.12), cropping enables the mask module to better exploit its foreground-background separation. This confirms that the components are complementary rather than redundant.
>
> | Setting          | MAE  | RMSE  |
> | ---------------- | ---- | ----- |
> | VM               | 7.14 | 14.44 |
> | VM+Crop          | 5.47 | 8.77  |
> | VM+Con+Crop      | 4.26 | 6.85  |
> | VM+Mask+Crop     | 4.12 | 5.94  |
> | VM+Mask+Con+Crop | 3.63 | 4.26  |
>
> ----
>
> **Q3: "Strong generalization" claim overstated; only two small-scale, structurally similar agricultural datasets.**
>
> We agree. We will revise "strong generalization to other agricultural scenes" to "consistent effectiveness across agricultural counting tasks with varying object densities and crop types" to better reflect the actual experimental scope.
>
> ----
>
> **Q4: Soft masking mechanism lacks formal detail; test split rationale and representativeness unclear.**
>
> (1) We will add the following formalization of soft masking in Section 3.2. In Stage 1, we compute a cosine similarity map $S$ between image patch embeddings and the text embedding, where $S_{ij} \in [-1, 1]$. In Stage 2, we apply per-sample min-max normalization to obtain $\hat{\mathbf{M}}$:
>
> $\hat{\mathbf{M}} = \frac{S - \min(S)}{\max(S) - \min(S)}$
>
> where $\hat{\mathbf{M}} \in [0,1]^{H \times W}$. The modulated input is then $\tilde{\mathbf{x}} = \mathbf{x} \odot \hat{\mathbf{M}}$, where $\odot$ denotes element-wise multiplication. Foreground regions (high similarity) are preserved while background regions (low similarity) are suppressed.
>
> (2) The train/val/test split was obtained by uniform random sampling, resulting in a representative test set (670 images, 71,690 total objects, 107 average count, 1,900 max count, 2.42‰ box ratio) whose statistics closely match the full dataset (6,265 images, 679,030 total objects, 108 average count, 1,980 max count, 2.38‰ box ratio), and we will include these details in the revised version.

---

### Official Review · Reviewer_EKPf · 2026-03-12

**Soundness:** 2
**Presentation:** 3
**Significance:** 3
**Originality:** 2
**Overall Recommendation:** 3
**Confidence:** 4

**Summary:**

This paper introduces the Dense and Indiscernible Object Counting (DIOC) problem, presenting a new dataset, DIOCblueberry, which contains 6,265 high-resolution images and over 679K annotations to benchmark counting in highly camouflaged and dense agricultural scenes. To tackle this, the authors propose MaskCount, a two-stage multimodal framework. Stage 1 utilizes Qwen3 to generate candidate background text prompts and employs CLIP to produce background-suppressing masks. Stage 2 estimates density maps using a frozen SwAV-pretrained ResNet50 backbone , augmented by an edge-aware patch cropping mechanism and a pixel-level contrastive loss.

**Compliance With Llm Reviewing Policy:**

Affirmed.

**Final Justification:**

I will maintain my original rating.

**Key Questions For Authors:**

1. How does the frozen ResNet50 backbone in Stage 2 handle the severe domain shift and artifacts introduced by the zero-value regions of the masked images? Have you evaluated the performance impact of unfreezing or fine-tuning this backbone?
2. In Table 8, why does a single Top-1 background prompt significantly outperform the Top-3 combination in highly complex agricultural scenes? Could you clarify if this indicates a potential vulnerability or mode collapse within the CLIP feature space for this specific task?
3. Relying on three manual exemplars per inference pass limits autonomous deployment. Have you evaluated modern Multimodal Large Language Models (MLLMs) directly in a zero-shot counting setting on the DIOCblueberry dataset?

**Limitations:**

Yes

**Strengths And Weaknesses:**

Strengths:
	Significance: The paper addresses a critical and highly practical bottleneck in precision 	agriculture. The curation of the DIOCblueberry dataset is a substantial contribution; the 	1,400 human hours invested in precise annotation yield a rigorous "hard mode" benchmark 	that pushes the boundaries of perception in low-discriminability environments.
	Presentation: The manuscript is structurally sound and well-written. The visual figures (e.g., 	the MaskCount pipeline in Figure 6 and mask visualizations in Figure 12) effectively 	communicate the challenges and the proposed solutions.
Weaknesses:
	Soundness: Feeding masked images with artificial zero-value regions into a frozen ResNet50 pre-trained on natural images introduces a potential domain shift. Additionally, extracting exemplar features via RoI Align from this masked output makes the model highly sensitive to first-stage segmentation errors.
	Originality: The methodological claims could be contextualized better. The proposed contrastive loss closely resembles standard cosine similarity in metric learning. Similarly, the edge-aware cropping shares core principles with the established overlap-tile strategy. Both are highly effective engineering adaptations rather than fundamental theoretical novelties.
	Significance: The multimodal design is somewhat limited, as Qwen3 is only used offline to generate a static background vocabulary. Furthermore, the method strictly requires three manual bounding boxes per image during inference. Exploring dynamic prompting or zero-shot capabilities would better align the work with current MLLM trends and autonomous agricultural deployment.

---

> ### Author Rebuttal · Authors · 2026-03-31
>
> **Comment:**
>
> Thanks a lot for your time and feedback. We have to say that the reviewer asks valuable questions and provides thoughtful clues. We appreciate your inspiring reviews. And we are happy to address the concerns.
>
> ---
>
> **Q1: Domain shift from zero-value masked input to frozen ResNet50; sensitivity to Stage-1 segmentation errors. (W1 & KQ1)**
>
> 1.  We would like to clarify that the mask does not introduce zero-value regions. **L241–247** in paper describes our soft masking strategy. The mask consists of continuous cosine similarity values in $[0,1]$, which modulate features via element-wise multiplication. We acknowledge that **Figure 6** may have caused this misunderstanding by depicting fully black masked regions. We will revise this figure to more accurately illustrate the soft masking mechanism.
> 2.  We conducted an additional experiment comparing **frozen vs. fine-tuned backbones under identical training configurations.** The frozen backbone clearly outperforms the fine-tuned variant, likely because fine-tuning on limited data disrupts well-learned general representations.
>
> | Setting    | MAE  | MSE  |
> | :--------- | :--- | :--- |
> | Fine-tuned | 5.87 | 9.08 |
> | Frozen     | 3.63 | 4.26 |
>
> 3.  We acknowledge that the mask accuracy in Stage 1 does have an impact on performance. However, **Table 4** demonstrates that the introduction of masks yields greater overall performance gains. In addition, we designed the **contrastive loss (Eq. 7)** to explicitly maximize the angular separation between foreground and background features. This mechanism compensates for imperfect mask boundaries by enforcing discriminative representations regardless of exact segmentation quality. The ablation in **Table 4** validates this.
>
> ---
>
> **Q2: The methodological claims could be contextualized better. e.g., Contrastive loss or edge-aware cropping (W2)**
>
> We appreciate this suggestion and will further clarify the applicability of our methods to the DIOC scenario.
>
> 1.  Our contrastive loss differs from standard metric learning in two ways:
>     *   (a) it operates at pixel level for foreground-background separation;
>     *   (b) background regions are dynamically determined by Stage-1 predictions.
>         In DIOC scenes where objects are densely packed and visually similar to the background, pixel-level separation is essential for accurate density estimation. Gradient analysis in **Appendix A.4** confirms reduced training variance. **Table 7** validates its superiority over alternative losses in DIOC.
> 2.  Unlike overlap-tile which fuses overlapping regions via averaging, we directly discard edge predictions and retain only the center area. In dense DIOC scenes, edge regions inevitably contain partial objects that cause double-counting. **Table 6** confirms the effectiveness of this strategy.
>
> ---
>
> **Q3: Zero-shot capability for MLLM. (W3 & KQ3)**
>
> This is an excellent point. We evaluated several MLLMs in a zero-shot counting setting on our DIOC dataset. We find:
>
> 1.  All MLLMs perform near random guessing on DIOC (best MAE 225.97 vs. ours 3.63), confirming that current MLLMs lack the capability for dense indiscernible object counting.
> 2.  We agree that dynamic prompting and zero-shot counting for DIOC are important future directions. We are very willing to study this issue in our future work.
>
> | Model             | MAE    | RMSE   |
> | :---------------- | :----- | :----- |
> | Qwen3.5-plus      | 247.17 | 464.53 |
> | InternVL3-78B     | 225.97 | 448.04 |
> | MiniCPM-V 4.5     | 261.51 | 490.15 |
> | GPT-4o-2024-05-13 | 254.91 | 493.27 |
>
> ---
>
> **Q4: Top-1 background prompt outperforms Top-3 in complex agricultural scenes; potential CLIP feature space vulnerability or mode collapse.(KQ2)**
>
> 1.  In complex agricultural scenes, CLIP **struggles to fuse attention maps from multiple background categories** (e.g., leaf and stem simultaneously). Top-3 masks under-cover complex backgrounds or over-cover target objects due to conflicting category descriptions. Top-1 retains only the highest-confidence background region for masking, producing cleaner separation.
> 2.  This is indeed a limitation of CLIP's fine-grained representation in DIOC agricultural scenes. CLIP is inherently stronger at single-category classification; when multiple background categories coexist, its discriminative ability degrades, explaining Top-3's inferior performance. We will discuss this limitation explicitly in the revised version.

---

> > ### Author Rebuttal · Reviewer_EKPf · 2026-04-04
> >
> > Thank you for the detailed rebuttal and the impressive additional experiments. The clarification on the soft mask and the ablation on the frozen vs. fine-tuned backbone effectively resolve my technical concerns regarding W1/KQ1. The candid admission of CLIP's limitations (KQ2) is also appreciated.
> >
> > I found the zero-shot MLLM evaluation (W3/KQ3) particularly interesting; it clearly demonstrates the current gap in large models for dense counting tasks. However, while your method achieves excellent performance, the strict requirement of three manual exemplars per inference pass remains a practical bottleneck for fully autonomous agricultural deployment. My technical concerns are mostly resolved, but I urge the authors to explicitly highlight this usability limitation—along with the new MLLM baseline results—in the final manuscript.

---

> > > ### Author Response · Authors · 2026-04-04
> > >
> > > Dear Reviewer EKPf,
> > >
> > > Thank you for your final endorsement and constructive feedback. **We will certainly formalize the zero-shot MLLM baseline results in the final manuscript.**
> > >
> > > We highly value your rigorous feedback on the usability bottleneck. As our primary contribution in this work is establishing the **DIOCblueberry benchmark**, we proposed **MaskCount primarily as a strong, standardized baseline to expose the challenges of this extreme domain**. Therefore, the current evaluation strictly adheres to the standard few-shot paradigm (providing exemplars per image) to ensure a fair and reproducible benchmark comparison.
> > >
> > > However, we completely agree that for real-world agricultural deployment, manual intervention must be minimized. In practice, a potential workaround is to cache and reuse the extracted appearance features across **subsequent images captured within the same crop environment**, though fully autonomous zero-shot initialization remains the ultimate goal.
> > >
> > > To transparently address this limitation, re-emphasize the role of our benchmark, and chart the path forward, we will add the following concise paragraph to **Section A.7 (Limitation Analysis)**:
> > >
> > > > *A limitation of MaskCount's few-shot paradigm is the reliance on three manual exemplars per image for initialization. While serving as a standardized baseline for the DIOCblueberry benchmark, we recognize that in practical autonomous deployment, exploring the caching and reusing of these exemplars across **subsequent images of the same category** could significantly minimize manual effort. Achieving fully zero-shot autonomous initialization remains a challenge. As demonstrated by our MLLM baselines, directly applying current large models without exemplars yields sub-optimal results due to the severe visual ambiguity of the DIOC task. Future work will build upon our dataset to explore eliminating this manual step entirely by leveraging MLLMs to autonomously generate the initial exemplars through a deeper understanding of the physical properties of objects in complex natural scenes.*
> > >
> > > We hope this accurately reflects our discussion and fully resolves your final request. Thank you again for your invaluable feedback!

---

### Decision · Program_Chairs · 2026-04-30

**Decision:**

Accept (regular)

**Comment:**

The paper receives one accept, two weak accepts, and one weak reject. After reviewing the rebuttal, all major concerns have been successfully resolved. The authors provided clarifying experiments on soft masks, backbone ablations, computational efficiency (12.9M parameters, single RTX 3090), and addressed concerns about MLP design and annotation costs. The remaining issue raised by the weak reject reviewer—the need for three manual exemplars per inference—is a practical limitation that the authors have agreed to explicitly highlight in the final manuscript. This does not undermine the technical contribution or the value of the DIOCblueberry benchmark. The work is technically solid, advances dense counting in agricultural domains, and meets ICML’s acceptance standard. I recommend acceptance.